# The Constrained-Disorder Principle Assists in Overcoming Significant Challenges in Digital Health: Moving from "Nice to Have" to Mandatory Systems

**Noa Hurvitz and Yaron Ilan \***

Hadassah Medical Center, Department of Medicine, Faculty of Medicine, Hebrew University, POB 1200, Jerusalem IL91120, Israel; noa.hurvitz@mail.huji.ac.il
\* Correspondence: ilan@hadassah.org.il

**Abstract:** The success of artificial intelligence depends on whether it can penetrate the boundaries of evidence-based medicine, the lack of policies, and the resistance of medical professionals to its use. The failure of digital health to meet expectations requires rethinking some of the challenges faced. We discuss some of the most significant challenges faced by patients, physicians, payers, pharmaceutical companies, and health systems in the digital world. The goal of healthcare systems is to improve outcomes. Assisting in diagnosing, collecting data, and simplifying processes is a "nice to have" tool, but it is not essential. Many of these systems have yet to be shown to improve outcomes. Current outcome-based expectations and economic constraints make "nice to have," "assists," and "ease processes" insufficient. Complex biological systems are defined by their inherent disorder, bounded by dynamic boundaries, as described by the constrained disorder principle (CDP). It provides a platform for correcting systems' malfunctions by regulating their degree of variability. A CDP-based second-generation artificial intelligence system provides solutions to some challenges digital health faces. Therapeutic interventions are held to improve outcomes with these systems. In addition to improving clinically meaningful endpoints, CDP-based second-generation algorithms ensure patient and physician engagement and reduce the health system's costs.

**Keywords:** digital health; artificial intelligence; variability; complex systems





## 1. Introduction

Artificial intelligence (AI) is still expected to influence healthcare delivery and the practice of medicine [1]. Despite the hype and attention around it and the rapid growth of digital technologies, the involvement of patients, clinicians, the insurance industry, and medicine regulators still needs to be higher]. The Valley of Death (VoD) is a challenge entrepreneurs, business owners, technology experts, innovators, and inventors must consider [2]. It reflects a series of challenges facing many companies in the digital world as the hype of the last decade seems to be over [2]. The "no evidence, no implementation–no implementation, no evidence" paradox is often related to the digital health field. The lack of evidence on how these systems may impact health outcomes, health systems efficiencies, and cost-effectiveness of service delivery is a significant challenge [3].

Digital health systems differ from other digital systems because patients do not approach health care voluntarily but because they are forced to. The challenge of implementing mandatory instead of "nice to have" systems poses a significant barrier to their implementation. Even though people in the field are beginning to believe that digital health has not met its expectations, that does not mean that it does not have a place; instead, it means rethinking some of the challenges. Whether AI succeeds in medicine and healthcare depends on its ability to penetrate the boundaries of evidence-based medicine, the lack of policies, and the reluctance of medical professionals to use it [4–7].

This paper discusses several challenges facing the digital field and describes using second-generation AI systems, which can answer some of them. Our paper emphasizes the need for mandatory platforms to improve outcomes rather than nice-to-have tools.

## 2. Uncertainty in the Healthcare Sector: Digital Health Has Failed to Meet Expectations

Uncertainty underlies the healthcare sector [8,9]. Patients sense there is no easy way to tap into the vast knowledge of healthcare services [10,11]. Healthcare is expensive, feels impersonal and corporate, and confronts people with multiple options and choices they need to prepare for [12]. The uncertainty is particularly pronounced in digital health, where many solutions still need to live up to their initial promise. Despite significant investments and efforts to implement digital technologies in healthcare, many solutions have yet to deliver the expected results [13]. Technological challenges, organizational barriers, and cultural resistance to change contribute to the failure of digital health to reach its expected potential [14]. A lack of education about the capabilities of digital medicine, and the added administrative burden that came with the early digitization of healthcare processes, contributed to physician burnout [15–17]. There is also fear that AI may eventually replace physicians [18]. The lack of a legal framework defining liability in adopting or rejecting algorithm recommendations leaves doctors vulnerable to legal consequences when using AI [19].

The healthcare community is repeatedly excited by the hope of providing better care through effective technology adoption [7,20–24]. There is no denying that digital health has not been delivered, and digital health has not transformed the health system. More healthcare intelligent technology (IT) companies have gone bankrupt in the past five years than in two decades before, and 98% of digital health startups have failed to survive [25,26]. Corporations are shutting down digital health labs, staunching investments in digital health, and consolidating digital health conferences, and governments are re-evaluating the funding regimes for such initiatives [27]. Digital health is yet to witness a large-scale adoption that could match the hope created about it [28,29].

Clinically robust, more inclusive, and better-personalized services are needed, using a marketplace model that incentivizes lower costs and better matching services with patients' needs [30].

## 3. Digital Health Trends over the Last Decade: First-Generation Systems

Many digital systems aim to achieve stand-alone digital interventions. To achieve this goal, therapeutic interventions must be integrated with digital systems, and drugs must complement digital-first interventions. Digital solutions are often grouped based on the potential risk to patients into solutions that improve system efficiency: measurable patient outcome benefit; mobile systems that inform or monitor and encourage change and self-management; and clinical decision support (CDS), and prediction models, that guide treatment, actively monitor, diagnose, and support treatment [7,31,32].

First-generation digital health tools were expected to resolve long-standing healthcare access and treatment inequalities in low- and middle-income countries [33]. However, many of these expectations were not met [34,35]. Expert systems can help diagnose medical conditions based on a patient's symptoms and other information. As a result of the patient's symptoms and other factors, the system would use a knowledge base of rules and information about different medical conditions to make a diagnosis. Moreover, they include systems that can assist with interpreting medical images, such as X-rays and MRIs, and systems that can analyze large amounts of data, such as electronic health records.

Monitoring is also part of the first-generation AI usage in medicine. Early detection of atrial fibrillation was one of the first uses of AI in medicine. Smartphone-based electrocardiogram (ECG) monitoring and detecting atrial fibrillation are regulatory approved [36]. While wearable and portable ECG technologies have helped detect atrial fibrillation, they have limitations, such as a high rate of false positives due to movement artifacts and

difficulty in adoption among elderly patients at higher risk [37]. With continuous glucose monitoring, people with diabetes can monitor their blood sugar levels in real-time. The FDA approved the Guardian glucose monitoring system. Sugar IQ is an app that helps users better prevent low blood sugar episodes based on repeated measurements [38], such as seizure detection with an alarm sent to close relatives and physicians [39]. AI-driven diagnostics perform cardiac ultrasound imaging without specialized training and systems that assist in radiology and pathology image interpretations [40–42]. Digital systems are proposed for improving precision medicine, such as matching genetic mutations found in tumor samples with patterns found in genetic data and medical records of patients [43,44]. The treatment can be personalized as a result.

Wearable biosensors can detect multiple cardiovascular parameters and multianalyte such as lactate, glucose, and electrolytes [45,46]. The application of health-tracking reward programs by insurance companies encourages using wearable health technology [47]. Decision support systems using signal and image processing improve diagnosing diseases in pathology and radiology [48]; telemedicine screening in ophthalmology assists in screening for glaucoma, diabetic retinopathies, and retinopathy of prematurity [49]. Autism and myocardial infarctions are also handled by telemedicine [50,51].

Risk stratification and prediction are also possible with this system [52,53], such as predicting the decline of glomerular filtration rate in patients with polycystic kidney disease [54] and risk stratification for the progression of IgA nephropathy [55]; and predicting outcomes in lower gastrointestinal bleeding [56], inflammatory bowel disease [57], esophageal cancer [58], and metastasis in colorectal cancer [59]. They can also be used for interpreting pulmonary function tests [60], processing images from endoscopy and ultrasound detecting abnormal structures in colonoscopy [61], and automate tedious, labor-intensive tasks, such as typing on keyboards and electronic records.

Telemedicine is used for follow-ups to reduce the cost of patients' accommodation in hospitals, providing access to doctors in rural areas [62]. It enables the implementation of health information systems considering countries' context, needs, vulnerabilities, and priorities [33]. Automated tools can fill gaps in trained healthcare workforce availability, simplify clinical workflows and assist in overcoming the lack of professionals [63]. In low-resource rural settings, the blockchain tracks supply chains in medication deliveries from pharmaceutical companies to hospitals and patients [64]. Thus, digital tools reduce inequalities in under-resourced neighborhoods, which are essential for promoting innovation and knowledge generation in low- and middle-income countries [33,65].

Using digital systems, treatment plans can be designed, and evidence-based treatment options provided by analyzing structured and unstructured data in medical records can be explored [66]. AI systems combine data from medical records with clinical expertise and research papers to suggest a treatment plan [4,67–69]. Using AI, a triage tool can identify high-risk patients and indicate their need for critical care early [70,71]. AI systems are used to facilitate drug development [72].

Virtual reality (VR)-based devices provide a virtual digital picture, while augmented reality (AR) results from integrating information or graphical elements into the user's environment in real time [73]. Mixed reality (MR), which combines AR and VR, improves the effectiveness of medical education and physician performance [74]. AR and VR are used to rehabilitate post-stroke and posttraumatic stress disorder [75].

The COVID-19 pandemic provided an opportunity to improve the collection and management of data to inform decision-making, screening, disease surveillance, and monitoring [76,77]. Digital health-assisted healthcare systems provide means for prevention and primary care [76,78]. Digital screening for COVID-19 decreased the number of visits to emergency departments [79]. Telemedicine for distance consultation and health gadgets, like pulse oximetry, were used in the COVID-19 pandemic [80,81].

Over the last decade, artificial intelligence has been used to analyze quantifiable data and perform highly repetitive tasks in healthcare [4]. Medical records can be collected, stored, and tracked digitally [82]. AI can improve in-person and online consultations

using a patient's medical history and common medical knowledge. A patient reports their symptoms through the app, which checks them against a database of diseases using speech recognition and offers suggestions [4,83,84]. As a virtual nurse, AI can assist patients with monitoring their health or managing diseases between doctor visits, providing health assistance and medication management [85]. Patients with chronic diseases can benefit from customized follow-up care, and treatment adherence is expected to increase [4,86–88].

Several smaller countries, such as South Korea and Estonia, have implemented digital health solutions, but their impact on the global population is small [89]. With the implementation of effective telemedicine and digital health record projects, the National Health Service (NHS) in England and Kaiser Permanente attempt to implement digital systems on a mass scale [90].

First-generation AI has had some success in the medical field, but its abilities are limited, and it cannot adapt to new situations or learn from its mistakes. Modern AI systems, including second-generation and beyond, are more flexible and can learn and adapt over time, which could make them more useful in medicine. To provide high-quality patient care, digital systems must provide a holistic approach that uses robust data and personalized parameters relevant to their healthcare needs [91].

Besides the direct application of artificial intelligence to medical data, generative AI, such as generative pre-trained transformers (GPT), is being introduced to enhance the precision, productivity, and clinical outcomes of patients [92]. Technologies like BERT (Bidirectional Encoder Representations from Transformers) and GPT comprise advanced natural language processing models with remarkable capabilities in interpreting and generating human-like text. Integrating BERT and GPT into digital health strategies may assist in information retrieval, patient–physician communication, and clinical decision-making. By leveraging their contextual understanding, these AI models can help physicians and patients sift through vast medical literature, decode complex terminology, and make informed choices about healthcare options [93–96]. In ophthalmology and radiology, various ways have been investigated to use ChatGPT-4 in research, medical education, and support for clinical decision-making [97]. There are, however, some limitations and risks associated with the current use of these systems [98]. GPT-like models are limited by the lack of domain expertise, the need to rely on the quality of the input data, the inability to detect errors, and the lack of understanding of ethical issues. Despite their ability to generate coherent and grammatically correct text, GPT-like models may lack the nuance and context that a human expert in the field would provide [98].

## 4. Challenges in Healthcare Systems That Need to Be Accounted for by Digital Systems

Healthcare markets are undergoing significant changes, and uncertainty, ambiguity, and complexity keeps increasing [99]. Healthcare systems worldwide are becoming unsustainable, and technology is asked to improve them [99–101], therefore the cost of care increases, including labor costs [102]. The staff shortage in healthcare systems is estimated to be 18 M by 2030 [103]. Reduced numbers of physicians are expected in many countries [104]. GPs have less than 15 min per patient, and 40% of emergency department visits are marked unnecessary [105]. Patient expectation increases and satisfaction in healthcare have been the lowest since 1997 [106]. Treatment must be tailored to the patient, and providers often fail to establish trust in the system.

The aging of the population means an increasing burden on healthcare services [107]. Diagnosis of several rare diseases takes 4–9 years [108]. Chronic conditions significantly burden healthcare systems, with 50% of GP appointments being for long-term conditions [109,110]. In total, 27% of health spending is on preventable illnesses [111]. There is low trust in pharma companies and their motives [112].

Pharma companies, payers, and health providers hesitate to use digital systems [113]. Though they "talk about it," they do not know how to implement these systems in a way that increases profits.

Health systems are conservative; however, the digital health world must respond fast to these challenges [114]. Changing processes require flexibility and adaptability and are often associated with costs [115]. Technology advances exponentially, but humans are linear, making it difficult to adapt to new systems. The narrative must shift from e-health/telemedicine to fitness devices, machine learning, artificial intelligence, blockchain, and automation [62,116,117]. Most healthcare providers fail to provide whole-person care. A patient-outcome-centered approach is more likely to succeed [118,119]. Automation is expected to reduce errors in healthcare and increase safety, while providing equity in care [120].

The following are a few challenges in developing a patient-tailored dynamic algorithm for diagnosis and treatment.

### 4.1. Digital Health's Data-Related Challenges with Machine Learning

Digital health relies extensively on mobile health, telemedicine, and various smart devices to collect human health data [78]. Even so, "big data" failed to translate into profitable products. Using machine learning (ML) in digital health poses several data-related challenges [121–126]. To use ML effectively, data must be high quality and complete. Nevertheless, digital health data can be challenging to collect and incomplete or noisy. When digital health data come from multiple sources, such as electronic health records, wearables, and mobile apps, heterogeneity and complexity can result. An ML model requires annotation and labeling of data. Particularly in digital health, where data may be complex and diverse, it can be time-consuming and labor-intensive. In terms of resources and infrastructure, storing and managing large amounts of digital health data can be challenging.

Potential algorithmic biases can occur if the data quality is unsatisfactory [127]. For AI to excel at tasks, it needs access to high-quality data [4,69,128]. Over the last decade, numerous systems have been associated with biases that evolve from the data type used [129]. Many current ML systems are based on existing data, not looking for relevant data [130,131]. It is common for algorithms to be trained using data from tertiary settings instead of primary care, which does not represent mild diseases well. It can have clinically significant effects, such as inflated case fatality estimates [132]. Overfitting training datasets and unforeseen errors from incidental variations or artifacts in input data can lead to bias in the algorithm's output [133]. The availability of real-world health data instead of the momentary snapshots seen in hospitals and clinics is mandatory for reforming disease management [134].

ML learns from historical data, and those underrepresented in these data sets may receive inaccurate diagnoses, so real-world data validation is crucial [132]. The population bias results from a focus on common presentations involving a solid predisposition toward training sub-populations [127,135,136]. Most data focus on subpopulations with limited ethnicity, such as white males, western populations with high income and literacy, and do not apply to many other subjects [127]. Algorithms often discriminate against women, minorities, other cultures, and ideologies. As such, algorithms learn from the data they are fed, and AI programmers must know about the issue of bias in algorithms to actively fight against it by tailoring them [137]. The information needs to account for differences in conditions in healthcare systems and how people are treated [100,138]. Building equitable sociodemographic representation in data repositories, gender, expertise, and clinical specialties is crucial in ameliorating health inequities [135]. There is a need to minimize dependence on trusted third parties or data movement [139]. When designing algorithms and introducing them into clinical practice, the principles of equity should be incorporated to ensure that the output does not cause harm to patients [33].

The democratization of data and care are mandatory [140]. Increasing amounts of data are being generated, which can be hard to account for, and interoperability requires sharing data and collaboration, which can be costly and require adaptability [141]. Data standardization, data access, overcoming biases due to limited datasets, efficient algorithm

deployment, and the need for data collaboration while keeping costs low are all barriers to scalable digital health [33]. More data is often unnecessary to improve its quality [142,143]. EMR (electronic medical records) are more than what may be required, and only some data and interactions are necessary [144]. Eighty percent of data remains silent as, in many cases, it is hard to take the data to the last mile toward the patient [145]. To provide meaningful information, the data must be translated into precision health [91]. It is possible for vulnerabilities, such as adversarial attacks and a lack of tools to regulate the quality of information and cybersecurity, to affect results negatively.

Commonly used systems focus on means of single points and are less accountable for the dynamicity of biological systems [146]. Using big data ignores the $n = 1$ concept, attempting to implement individual conclusions made from large populations, which may result in a bias that affects the treatment of patients [146–148].

### 4.2. Patients and Physicians Face Challenges in Using Digital Systems

Most digital systems need to engage patients or physicians. In addition, physicians are reluctant to use ML platforms because patients ignore them [69,149]. Despite expectations, healthcare players still needed to adopt digital strategies. Providers must convince patients of the benefits of using a new system, so explainable AI is crucial. For most patients and providers, the system must be relevant to the moment rather than the future. Innovations are accepted at different levels by end users [150,151]. The challenge of attitude comes from changing mindsets. For a system to be used for a long time, the user must want to use it.

It is possible that these systems need to be better understood or trusted, especially if they are perceived as complex or unfamiliar. Patients and physicians may only accept AI if they fully understand how it works and are sure about its reliability and accuracy. With physicians already having varying technology literacy levels, frustrations may be added as physicians learn how to incorporate and utilize AI platforms while struggling with existing technologies such as EMR [152,153]. Physician burnout can be exacerbated by understanding how AI algorithms work. As a result of AI, there are concerns about the potential for errors or biases and the possibility that these systems replace human interaction and support. It is particularly relevant for patients who may be concerned about the accuracy and reliability of AI-based diagnoses or treatments [154,155], and how these systems may undermine healthcare professionals' autonomy and judgment [156].

Augmented intelligence implies the AI's assistive role by enhancing human intelligence rather than replacing it [1,157]. In addition, it refers to combining the unique capabilities of human experts with AI to provide better care [4]. Medical professionals make decisions using data obtained with technologies they understand [158]. An explainable AI is crucial to gaining trust in AI-based algorithms [158,159]. New technologies tend to be accepted by physicians if they add to their knowledge of diagnosis and treatment, increase income, and save time by allowing them to do more in their practice. Knowledge, money, and time are the underlying benefits of investing in digital health [69,160,161].

Usability testing examines whether specified users can achieve their intended use effectively and efficiently [162,163]. The VoD often occurs during the clinical translation stage of digital tools due to issues with AI performance, generalizability, black boxes, and explainability [33].

A patient should be involved in the highest decision-making level when designing algorithms for medical purposes to ensure their needs and recommendations are implemented [4,164]. Overall, 75% of consumers want a more personalized experience [165,166]. Balancing technology with real people is needed.

GPT and BERT models lack domain-specific medical knowledge and may generate plausible-sounding yet medically inaccurate information. Integrating medical expertise into these models' training and fine-tuning processes is essential for reliable medical applications. Furthermore, medical language is highly technical and filled with domain-specific jargon. GPT and BERT may struggle to handle the linguistic complexity and diverse terminology in medical literature and patient records [167].

Patients and medical professionals need time and resources to trust AI with medical diagnoses, support medical decision-making, or design therapies [4]. Trust is associated with a system's performance and ability to improve outcomes. Scaling innovations for increasing adoption by clinician- and patient-mandate large investments is a significant challenge [168,169].

### 4.3. Challenges Related to Ethics and Law

Liability needs to be resolved if AI systems cause errors or adverse outcomes. Who is responsible for the consequences if an AI system provides an incorrect diagnosis or treatment recommendation? Several parties are involved, including the developers of the AI system, the healthcare providers who use the system, and the affected patients [168,169]. If an algorithm misses a diagnosis that a physician accepted, the consensus is that the professional is liable if the tool was misused [170]. In other cases, the liability falls back on the creators and the companies behind them [4]. With AI and ML, human doctors are subjected to confirmation bias. The system often tends to reproduce doctors' errors, thus strengthening their mistakes [171,172].

Due to legal concerns, data with personal identifiers may not always be distinguishable from fully anonymous data. If the data provider does not trust the potential recipient, they may be reluctant to share their data. Official guidelines on data sharing may be absent or unclear, making it difficult to determine how to balance data accessibility with the need to protect privacy and intellectual, financial, and time investments.

GPT- or BERT-generated information might inadvertently lead to incorrect diagnoses, inappropriate treatment recommendations, or miscommunication between healthcare providers and patients [167,173].

Fairness in digital applications requires ethical considerations. A fair selection process considers differences in race, gender, demographic disparities, disability, and other factors [33]. There may also be ethical concerns, as public health agencies may disagree with data requestors about the risks and benefits of sharing data. Additionally, data producers may feel they need more credit or benefit in transit, where they share their data, while data users may benefit from academic credit and career advancement [174].

It is challenging to own and access data generated by AI systems. It may need to be clarified who owns and has access to the data. There may also be concerns about the privacy and security of this data, as it often contains sensitive personal information [175]. The COVID-19 pandemic revealed the need for data sharing and for evaluation and ethical aspects to be developed in the emerging field of digital healthcare, such as consent and transparency regarding what data are collected, and which third parties can access patient data [78,176].

In many cases, it has been proven that individual profiles can still be traced back even if data is anonymized by institutions [177]. Ethical challenges, such as user consent, are significant in health digitalization [178]. Data governance is another challenge, and governments must set up policies and standards for data governance [179].

### 4.4. Challenges Related to Healthcare Providers and Pharmaceutical Companies

Digital solution evaluation and adoption requires collective efforts from multiple parties, such as health authorities, healthcare providers, manufacturers, small and medium-sized enterprises (SMEs), and multinational corporations (MNCs) [7]. More giant corporations have more resources to develop evidence, but are equally limited by time and have much hesitancy to use digital platforms [4,180]. It can be challenging to justify investments into expensive and time-consuming clinical studies for early stage solutions for internal budget allocation. The evidence published today may reflect a product that has been updated and refined multiple times since investigations typically take three years. Sales and manufacturing investments are more tangible for companies, with a more predictable return on investment than clinical studies [7,181]. Researchers may not be willing to conduct studies to evaluate digital solutions, which require different settings and capabilities and

whose outputs involve benefits on the operational and cost level, and therefore, indirectly, patient outcome versus a drug that improves patient outcome is a challenge [182].

### 4.5. Cost-Increase Challenges in Healthcare

The cost-effectiveness and sustainability of digital solutions are significant challenges for implementing these systems [78,183,184]. AI systems pose a significant barrier to implementation in most countries due to their burden on healthcare systems [185]. In parallel, there is a need for value-based health care in low-income places [186].

Implementing AI systems in healthcare requires a significant initial investment, including the costs of acquiring or developing the technology, training staff, and integrating the systems with existing information technology infrastructure. It increases costs in the short term [187]. AI systems require ongoing maintenance and support, which incurs additional costs. Software updates, hardware repairs, staff training, and data storage and processing may all be necessary to ensure the smooth operation of AI systems [188,189]. Complying with relevant laws and regulations, such as HIPAA, involves additional costs, such as legal fees and registration fines.

Digital platforms are asked to assist in treating more patients with the same personnel and require readiness of local infrastructures such as clinical services, equipment, treatment modalities, IT systems, telecommunication network, and cost of AI platforms [190,191]. Resource limitations, workforce, and infrastructure are presently significant barriers to the translation of benefits of digital technologies to improve public health measures, particularly in low- to middle-income countries [192–194]. Most payers and healthcare systems do not justify adding costs for digital systems, which are "nice to have".

Unlike drugs, where the evidence leads to reimbursement, reimbursement for digital systems requires showing their value and potential in the real world, not just in clinical trials. To receive reimbursement, digital systems must be affordable and cost-saving. Most digital apps address problems that seem "not critical"; they do not improve survival, making it challenging to ask health systems for reimbursement.

### 4.6. Regulations, Validations, and Standards Challenges

The regulatory process for medical devices includes establishment registration, listing, and premarket notification or approval. The process is very complicated and lengthy, and The FDA acknowledges that traditional forms of medical-device regulation are not well suited for the faster pace of design and modification for software-based medical devices [195].

Numerous regulations and standards are relevant to digital health solutions [7,24,196]. Regulations must provide life cycle requirements for developing medical software, such as that found within medical devices, and communicating electronic health record (EHR) information [197]. These instruct on the principles and requirements for privacy protection using pseudonymization services to protect personal health information [7]. Standards and criteria are defined in some regulations for ensuring the interoperability of components used in applications monitoring personal health and wellness [7]. Guidelines provide frameworks for evaluating the benefits and risks of digital solutions, guidelines on effectiveness, and economic standards [198–200].

The challenges of interoperability, data privacy, legal frameworks, systemic acceptability, and project financing are obstacles to large-scale digital platforms [201–205].

While the strength of evidence and study duration is mandatory for proper assessment of the efficacy of digital systems, only a limited number of products were tested in RCT [206]. Examples of studies are patients randomly assigned to routine outpatient chemotherapy for advanced tumors with patient-reported outcomes vs. usual care with monitoring at the discretion of clinicians [207]; text messaging to reduce early discontinuation of aromatase inhibitor therapy in breast cancer [208]; patients with type 2 diabetes using cell phone-based software [209]; clinical decision support system to aid computerized orders entry of

chemotherapy [210]; and the use of a deep-learning framework (Med3R), which utilizes human-like learning and reasoning process [7,211].

For evaluating digital health solutions, the pre-post design is most commonly used. It involves pre-phase, which provides control data; a "washout" period with no interventions implemented with a time gap of up to several months to allow familiarization and to limit bias related to implementation and post-phase to collect data on solution effectiveness [212]. There are differences between the evidence required for initial adopters (e.g., surveys and interviews, case studies) and those needed for the majority (prospective RCT studies) [213–215].

Before implementing digital systems, RCTs are required, which is challenging and may delay their implementation by years at a time when technology has already advanced. The value of digital systems may be better reflected in real-world studies. Collaboration between companies can benefit many systems, but it is difficult to implement and encourage.

## 5. Moving from "Nice to Have" to "Mandatory" Digital Systems

Several challenges, including those listed above, present significant obstacles to implementing digital systems. While many believe time and fine-tuning of these systems can lead to breaching everyday life, the reality is more complicated. Most digital systems are still considered "nice to have" and are not required for care.

The purpose of healthcare systems is to improve outcomes. Diagnoses, data collection, and simplifying processes are "nice to have" tools but not mandatory. The majority of these systems have never been shown to improve outcomes. "Nice to have", "assist", and "ease processes" do not suffice in the current outcome-based environment. Systems need to adapt fast to changing circumstances [216]. Most current systems are not sufficiently dynamic in response to internal and external perturbations [217]. There is a need to be both fast and accurate, make all actions computable, and support different data types, algorithms, and statistics for accounting for the dynamicity of biological systems [218]. A challenge for end users, such as patients and providers, is determining a new solution's credibility and compliance with standards [7,33].

Unless a digital platform improves outcomes, it is unlikely to break through the glass ceiling of everyday use in healthcare. The lack of resources makes digital systems mandatory for improving outcomes, ensuring a high engagement rate by patients and providers, and being reimbursed.

## 6. Constrained-Disorder Principle-Based Digital Systems Get Closer to Their Biological Basis

The disorder is inherent to the function of complex systems, and variability characterizes the proper operation of biological systems [219–222]. At the genome, a combination of deterministic and stochastic effects regulate processes of DNA transcription and translation [223]. At the cellular level, dynamic instability is the hallmark of normal microtubule function [224–230]. Heart rate variability (HRV) and blood pressure variability (BPV) are examples of an autonomic nervous system, where normal regulation of the heart and vascular function happens [231]. Loss of HRV is associated with poor prognosis and increased mortality [232,233]. Abnormal BPV affects the morphology and composition of coronary plaques and the related mechanisms of inflammation and hemodynamics. Regulating BPV can prevent the occurrence and development of coronary heart disease [234].

The constrained-disorder principle (CDP) defines complex systems based on the degree of their inherent disorder, bounded by dynamic boundaries [235]. It differentiates between living organisms characterized by a high degree of disorder and non-living systems with a minimal degree. A system's malfunction evolves from reducing the degree of disorder or when disorder becomes out of bounds [235–238]

CDP-based second-generation AI systems are designed to generate therapeutic regimens close to human biology, improving the response to chronic therapies, and thus clinical outcomes [146]. By using personalized algorithms, these systems incorporate controlled

variability into therapeutic interventions. Chronic diseases are a significant burden on healthcare systems, and the loss of response to chronic drugs is a significant challenge in treating patients with common chronic diseases [239]. Patients with chronic diseases also fail to adhere to chronic regimens because of a loss of response to therapies. In addition, there is marked inter and intra-subject variability in response to chronic therapies [239–242]. The CDP-based second-generation AI systems provide a platform for overcoming drug resistance and improving adherence by implementing variability-based therapeutic regimens for patients with chronic diseases [76,108,242–260]. The system enables personalized therapies based on individual variability signatures [146,239,257,261].

The CDP-based second-generation AI is a platform on which the digital pill was developed [261]. The digital pill is any drug regulated by a second-generation AI and is available at three levels. First, an open-loop system implements variability in drug administration times and dosages to overcome tolerance. At the second level, a closed-loop design personalized the variability signatures in a way that dynamically adapts the variability to each subject's response. At the third level, the algorithm incorporates signatures of variability, which are relevant to the disease dynamically [245,258,261]. Examples are the use of HRV in cardiac patients, quantifying variability in cytokines secretion in patients with inflammatory disorders, and respiratory and gait variability parameters in patients with pulmonary disease and neurological disorders [146,239,257,261]. With this CDP-based digital system, patients with severe heart failure had fewer emergency room visits, fewer hospitalizations, improved clinical performance, and improved laboratory tests.

By viewing the digital pill as part of therapy, not as a reminder, patients and physicians are more likely to engage with the platform. It provides end users with confidence that their outcomes will improve if they take their medications according to the app-based regimens. When introducing digital systems, healthcare organizations, payers, and end users incur high costs. The digital pill is based on an "Uber/Airbnb" model, where a digital system improves the efficiency and effectiveness of existing drugs and devices [245,258,261,262].

## 7. Digital Health Challenges Can Be Overcome by Using CDP-Based Digital Systems

With CDP-based digital systems, patient outcomes are improved; therefore, some challenges associated with digital health are overcome.

Since patients are the "kings of healthcare," digital health must transform from a stand-alone product to a service that supports clinical outcomes [76,108,217,243–260,263–267]. Unlike first-generation systems, second-generation AI systems are outcome-based, with clinically meaningful endpoints. Healthcare players can rely on them for solutions and to add value to their systems. Patients benefit from them because they improve their response and reduce side effects. They focus on the patient's essential endpoints to overcome the challenge of technology-patient interaction. Healthcare providers ensure improved adherence. Payers reduce costs by reducing admissions and the need for expensive therapies that are not necessarily better. Pharma companies can increase revenues without developing new expensive products or dealing with regulatory barriers using these methods [261].

These systems improved clinical and laboratory measures in patients with chronic heart failure, reducing emergency room admissions and hospitalizations [268]. Similar results were shown for patients with multiple sclerosis and those suffering from chronic pain [238]. These examples support the concept that the introduction of outcome-based digital systems can overcome many of the above-discussed challenges.

In many cases, big data can be overcome using second-generation systems, which is of limited relevance when designing therapies for individual subjects. By generating insightful datasets, the systems create individualized parameters associated with drug effectiveness, adherence, and side effects [261,262].

Second-generation systems must dynamically retune biological, environmental, and social factors [33]. They are dynamic and continuously change their output based on internal and external perturbations. By improving outcomes, these systems are turning "nice to have" digital algorithms into mandatory ones [262].

Table 1 summarizes some of the challenges and the suggested methods for overcoming them using the described system.

**Table 1.** Some challenges faced by digital health systems and suggested methods for overcoming them using CDP-based systems.

| | **Digital Health System Challenge** | **Constrained-Disorder Principle-Based Second-Generation Artificial Intelligence Solutions** |
|---|---|---|
| **Data** | "Big data" failed to translate into improving patient outcome | Generating insightful, personalized datasets for subject-tailored therapeutic regimens [146,236,239,261,262] |
| **Users** | Lack of engagement by patients and physicians | Outcome-based systems ensure long-term engagement as patients view the system as part of the therapy [262]. |
| | A need for explainable systems | The improved outcome is quantifiable in most cases, easing the process of adapting to digital systems [146,261,262] |
| **System functions** | Biases | The system reduces biases as it is independent of the physician. Algorithms are targeted to clinically meaningful outcomes [146]. |
| **Payers** | Increased costs | By improving outcomes, the system reduces hospitalizations and the need for more expensive therapies, thus saving costs [261]. |
| **Pharma companies** | Cannot translate digital system to profits | Improving adherence increases sales while providing pharma with a market disruptor [261]. |
| **Validation** | Difficulty in validating advantages | Outcome-based endpoints are quantifiable and, in most cases, are easily validated [238,268]. |

## 8. Summary and Conclusions

In the case of digital health, it can either be declared dead or resurrected [269,270]. To move digital platforms into everyday use, ensure high engagement by patients and physicians, and ensure reimbursement, it is crucial to differentiate between "nice to have" systems and mandatory systems. Any new technology needs to be pragmatic, solve problems, reduce the cost of care delivery, and be sustainable in the long term [5,78]. As long as it benefits players in the health system, doing so one step at a time is reasonable as long as it does not require perfection. A CDP-based second-generation AI system is showing promise and has the potential to overcome some of the challenges digital systems face. As long as medical associations provide clear guidelines for implementing AI and policymakers create policies encouraging its adoption, AI can become part of standard care [4]. It all depends on the ability to show improved outcomes when using digital systems, moving from "nice to have" into mandatory systems.

**Author Contributions:** N.H. and Y.I. wrote the manuscript. Y.I. conceptualized. All authors have read and agreed to the published version of the manuscript.

**Funding:** This research received no external funding.

**Institutional Review Board Statement:** Not applicable.

**Informed Consent Statement:** Not applicable.

**Data Availability Statement:** All data is available on public domains.

**Conflicts of Interest:** The authors declare no conflict of interest.

## Abbreviations

AI: artificial intelligence; CDP: constrained disorder principle; VoD: valley of death; IT: intelligent technology; CDS: clinical decision support; MRI: magnetic resonance imaging; ECG: electrocardiogram; FDA: food and drug administration; BERT: Bidirectional Encoder Representations from Transformers; GPT: Generative pre-trained transformers;

VR: Virtual reality; AR: augmented reality; MR: Mixed reality; NHS: National Health Service; GP: General practitioner; ML: machine learning; EMR: Electronic medical records; SMEs: small and medium-sized enterprises; MNCs: multinational corporations; HIPAA: Health Insurance Portability and Accountability Act; EHR: electronic health record; RCT: randomized controlled trail; HRT: heart rate variability; BPV: blood pressure variability.

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
