# Peer review of "The Constrained-Disorder Principle Assists in Overcoming Significant Challenges in Digital Health: Moving from “Nice to Have” to Mandatory Systems"

_clinpract, doi:10.3390/clinpract13040089_

Round 1
Reviewer 1 Report
I am really grateful to review this manuscript. In my opinion, this manuscript can be published once some revision is done successfully. I made one suggestion and I would like to ask your kind understanding. This study used 264 references to review the role of second-generation artificial intelligence based on constrained-disorder principle (CDP) in overcoming significant challenges in digital health such as increasing uncertainty and cost. I would argue that this is a good start. However, it can be noted that this study covered only machine learning and missed a recent emergence of generative artificial intelligence such as BERT and GPT. I would like to ask the authors to address this issue througout the manuscript.
Minor editing of English language required
Author Response
I am really grateful to review this manuscript. In my opinion, this manuscript can be published once some revision is done successfully. I made one suggestion and I would like to ask your kind understanding. This study used 264 references to review the role of second-generation artificial intelligence based on constrained-disorder principle (CDP) in overcoming significant challenges in digital health such as increasing uncertainty and cost. I would argue that this is a good start. However, it can be noted that this study covered only machine learning and missed a recent emergence of generative artificial intelligence such as BERT and GPT. I would like to ask the authors to address this issue througout the manuscript.
Response: The authors accept the comment. We have added a section on generative artificial intelligence, including BERT and GPT, as suggested. The relevant references were included.
Reviewer 2 Report
The paper presents a review of the systems designed for digital health systems.
The abstract should cover the purpose, a description of the review relevant to the state-of-the-art methods, and a brief overview of the future perspective of digital health systems.
Different sections cover the details of different methods included under different headings. A review article should cover the problems faced by each category of the systems. There must be some comparative analysis of different methods. Only theoretical discussion on different methods does not make a review article useful for the readers. Half of the article consists of the bibliography.
Column 3 in Table 1 needs to be supported by references to some available solutions designed to tackle the challenge.
There should be a conclusion section with future research directions in digital health systems.
Author Response
The paper presents a review of the systems designed for digital health systems.
The abstract should cover the purpose, a description of the review relevant to the state-of-the-art methods, and a brief overview of the future perspective of digital health systems.
Response: The authors accept the comment. The abstract was revised as suggested.
Different sections cover the details of different methods included under different headings. A review article should cover the problems faced by each category of the systems. There must be some comparative analysis of different methods. Only theoretical discussion on different methods does not make a review article useful for the readers. Half of the article consists of the bibliography.
Response: We accept the remark. We have revised the relevant section to highlight the main problems in each category. We have also broadened the sections discussing the potential solutions providing examples for improved clarity. As this is a review, we attempted to include the relevant publications.
Column 3 in Table 1 needs to be supported by references to some available solutions designed to tackle the challenge.
Response: Corrected.
There should be a conclusion section with future research directions in digital health systems.
Response: A conclusion section was included as suggested.
Round 2
Reviewer 2 Report
Most of my concerns are addresses in the revised version.